# LEARNING REPRESENTATIONS IN MODEL-FREE HIERARCHICAL REINFORCEMENT LEARNING

## ABSTRACT

Common approaches to Reinforcement Learning (RL) are seriously challenged by large-scale applications involving huge state spaces and sparse delayed reward feedback. Hierarchical Reinforcement Learning (HRL) methods attempt to address this scalability issue by learning action selection policies at multiple levels of temporal abstraction. Abstraction can be had by identifying a relatively small set of states that are likely to be useful as subgoals, in concert with the learning of corresponding skill policies to achieve those subgoals. Many approaches to *subgoal discovery* in HRL depend on the analysis of a model of the environment, but the need to learn such a model introduces its own problems of scale. Once subgoals are identified, skills may be learned through *intrinsic motivation*, introducing an internal reward signal marking subgoal attainment. In this paper, we present a novel model-free method for subgoal discovery using incremental unsupervised learning over a small memory of the most recent experiences of the agent. When combined with an intrinsic motivation learning mechanism, this method learns subgoals and skills together, based on experiences in the environment. Thus, we offer an original approach to HRL that does not require the acquisition of a model of the environment, suitable for large-scale applications. We demonstrate the efficiency of our method on two RL problems with sparse delayed feedback: a variant of the rooms environment and the ATARI 2600 game called *Montezuma's Revenge*.

## 1 INTRODUCTION

The reinforcement learning problem suffers from serious scaling issues. *Hierarchical Reinforcement Learning* (HRL) is an important computational approach intended to tackle problems of scale by learning to operate over different levels of *temporal abstraction* (Sutton et al., 1999). The acquisition of hierarchies of reusable skills is one of the distinguishing characteristics of biological intelligence, and the learning of such hierarchies is an important open problem in computational reinforcement learning.

A number of general approaches have been suggested towards this end. One approach focuses on action sequences, subpolicies, or "options" that appear repeatedly during the learning of a set of tasks. Such frequently reused subpolicies can be abstracted into skills that can be treated as individual actions at a higher level of abstraction. A somewhat different approach to temporal abstraction involves identifying a set of states that make for useful *subgoals*. This introduces a major open problem in HRL: that of *subgoal discovery*.

A variety of researchers have proposed approaches to identifying useful subpolicies and reifying them as skills (Pickett & Barto, 2002; Thrun & Schwartz, 1995). For example, Sutton et al. (1999) proposed the *options framework*, which involves abstractions over the space of actions. At each step, the agent chooses either a one-step "primitive" action or a "multi-step" action policy (an option). Each option defines a policy over actions (either primitive or other options) and comes to completion according to a termination condition.

Other researchers have focused on identifying subgoals — states that are generally useful to attain — and learning a collection of skills that allow the agent to efficiently reach those subgoals. Some approaches to subgoal discovery maintain the value function in a large look-up table (Sutton et al., 1999; Goel & Huber, 2003; Şimşek et al., 2005), and most of these methods require building the

state transition graph, providing a model of the environment and the agents possible interactions with it (Machado et al., 2017; Şimşek et al., 2005; Goel & Huber, 2003). Formally, the state transition graph is a directed graph $G = (V, E)$ with a set of vertices, $V \subseteq \mathcal{S}$ and set of edges $E \subseteq \mathcal{A}(\mathcal{S})$, where $\mathcal{S}$ is the set of states and $\mathcal{A}(\mathcal{S})$ is the set of allowable actions. Since actions typically modify the state of the agent, each directed edge, $(s, s') \in E$, indicates an action that takes the agent from state $s$ to state $s'$. In nondeterministic environments, a probability distribution over subsequent states, given the current state and an action, $p(s'|s, a)$, is maintained as part of the model of the environment. Previously proposed subgoal discovery methods have provided useful insights and have been demonstrated to improve learning on relatively small tasks, there continue to be challenges with regard to scalability and generalization. Scaling to large state spaces will generally mandate the use of some form of nonlinear function approximator to encode the value function, rather than a look-up table. More importantly, as the scale of reinforcement learning problem increases, the tractability of obtaining a good model of the environment, capturing all relevant state transition probabilities, precipitously decreases.

Once useful subgoals are discovered, an HRL agent should be able to learn the skills to attain those subgoals through the use of *intrinsic motivation* — artificially rewarding the agent for attaining selected subgoals. The nature and origin of "good" intrinsic reward functions is an open question in reinforcement learning, however, and a number of approaches have been proposed. Singh et al. (2010) explored agents with intrinsic reward structures in order to learn generic options that can apply to a wide variety of tasks. Value functions have also been generalized to consider goals along with states (Vezhnevets et al., 2017). Such a parameterized universal value function, $q(s, a, g; w)$, integrates the value functions for multiple skills into a single function approximator taking the current subgoal, $g$, as an argument.

Recently, Kulkarni et al. (2016) proposed a scheme for temporal abstraction that involves simultaneously learning options and a control policy to compose options in a deep reinforcement learning setting. Their approach does not use separate $Q$-functions for each option, but instead treats the option as part of the input. However, the method of Kulkarni et al. (2016) does not include a technique for automatic subgoal discovery, forcing the system designer to specify a set of promising subgoal candidates in advance. The approach proposed in this paper is inspired by Kulkarni et al. (2016), which has advantages in terms of scalability and generalization, but incorporates automatic subgoal discovery.

It is important to note that *model-free* HRL, which does not require a model of the environment, still often requires the learning of useful internal representations of states. When learning the value function using a nonlinear function approximator, such as a deep neural network, relevant features of states must be extracted in order to support generalization at scale. A number of methods have been explored for learning such internal representations during model-free reinforcement learning Tesauro (1995); Rafati & Noelle (2017); Mnih et al. (2015).

In this paper, we seek to address major open problems in the integration of internal representation learning, temporal abstraction, automatic subgoal discovery, and intrinsic motivation learning, all within the model-free HRL framework. We propose and implement efficient and general methods for subgoal discovery using unsupervised learning methods – such as $K$-means clustering and anomaly (outlier) detection. These methods do not require information beyond that which is typically collected by the agent during model-free reinforcement learning, such as a small memory of recent experiences.

Our methods are fundamentally constrained in three ways, by design. First, we remain faithful to a model-free reinforcement learning framework, eschewing any approach that requires the learning or use of an environment model. Second, we are devoted to integrating subgoal discovery with intrinsic motivation learning. Specifically, we conjecture that intrinsic motivation learning can increase appropriate state space coverage, supporting more efficient subgoal discovery. Lastly, we focus on subgoal discovery algorithms that are likely to scale to large reinforcement learning tasks. The result is a unified algorithm that incorporates the learning of useful internal representations of states, automatic subgoal discovery, intrinsic motivation learning of skills, and the learning of subgoal selection by a "meta-controller", all within the model-free hierarchical reinforcement learning framework. We demonstrate the effectiveness of this algorithm by applying it to a variant of the rooms task (illustrated in Figure 2(a)), as well as a classic and difficult ATARI 2600 game called *Montezuma's Revenge* (illustrated in Figure 3(a)).

## 2    REINFORCEMENT LEARNING PROBLEM

The Reinforcement Learning (RL) problem is learning through interaction with an *environment* Sutton & Barto (1998). At any time step the *agent* receives a representation of the environment's *state*, $s \in \mathcal{S}$, where $\mathcal{S}$ is the set of all possible states, and, on that basis, the agent selects an *action*, $a \in \mathcal{A}$, where $\mathcal{A}$ is the set of all available actions. One time step later as a consequence of the agent's action, the agent receives a *reward* $r \in \mathbb{R}$ and also an update on the agent's new state, $s'$, from the environment. Each cycle of interaction is called a transition *experience*, $e = (s, a, r, s')$. At each time step, the agent implements a mapping from states to possible actions, $\pi : \mathcal{S} \rightarrow \mathcal{A}$, called its *policy*. The goal of the RL agent is to find an *optimal policy* that maximizes the expected value of the *return*, i.e. the cumulative sum of future rewards, $G_t = \sum_{t'=t}^{T} \gamma^{t'-t} r_{t'+1}$, where $\gamma \in (0, 1]$ is the *discounted factor* and $T$ is a final step. The Temporal Difference (TD) learning approach is a class of *model-free* RL methods that attempt to learn a policy without learning a model of the environment. It is often useful to define a *value* function $q_\pi : \mathcal{S} \times \mathcal{A} \rightarrow \mathbb{R}$ to estimate the expected value of the return, following policy $\pi$. When the state space is large, or not all states are observable, we can use a function approximator $Q(s, a; w)$, such as an artificial neural network, to estimate the value function $q_\pi$. Q-learning is a TD algorithm that attempts to find the optimal value function by minimizing the loss function $L(w)$, which is defined as the expectation of squared *TD error* over a recent transition *experience memory* $\mathcal{D}$:

$$L(w) \triangleq \mathbb{E}_{(s,a,r,s') \sim \mathcal{D}} \Big[ \big( r + \gamma \max_{a'} Q(s', a'; w) - Q(s, a; w) \big)^2 \Big].$$

## 3    A UNIFIED MODEL-FREE HRL FRAMEWORK

In Hierarchical Reinforcement Learning (HRL), a central goal is to support the learning of representations at multiple levels of abstraction. As a simple example, consider the task of navigation in the *4-room environment with a key and a lock* in Figure 2(a). This is a variant of the *rooms* task introduced by Sutton et al. (1999). The 4-room is a grid-world environment consisted of 4 rooms. Each grid square is a state, and the agent has access to the Cartesian location of each grid square. Actions allow the agent to move to an adjacent grid square. The 4 rooms are connected through *doorways*. The agent is rewarded for entering the grid square containing the key, and it is more substantially rewarded for entering the grid square with the lock after obtaining the key. Learning this task based on sparse delayed feedback is challenging for a reinforcement learning agent.

Our intuition, shared with other researchers, is that hierarchies of abstraction will be critical for successfully solving problems of this kind. To be successful, the agent should represent knowledge at multiple levels of spatial and temporal abstraction. Appropriate abstraction can be had by identifying a relatively small set of states that are likely to be useful as *subgoals* and jointly learning the corresponding skills of achieving these subgoals, using intrinsic motivation.

In this section, we introduce a unified method for model-free HRL. The major components of our framework, and the information flow between them, are sketched in Figure 1(a). Before describing the unified method, we introduce the various components of our framework.

### 3.1    META-CONTROLLER AND CONTROLLER FRAMEWORK

Inspired by Kulkarni et al. (2016), we use two levels of hierarchy for learning internal representations for value function approximation. The more abstract level of this hierarchy is managed by a *meta-controller* and this system guides the action selection processes of the *controller*. Separate value functions are learned for the meta-controller and the controller, but they jointly learned, together. The process is illustrated in Figure 1(b). At time step $t$, the meta-controller that receives a state observation, $s = s_t$, from the environment. It has a policy for selecting a *subgoal*, $g = g_t$, from a set of subgoals, $\mathcal{G}$. In our implementation, the policy arises from estimating the value of each subgoal, $Q(s, g; \mathcal{W})$, and selecting the goal of highest estimated value (except when performing random exploration). With the current subgoal selected, the controller uses its policy to select an action, $a \in \mathcal{A}$, based on the current state, $s$, and the current subgoal, $g$.

In our implementation, this policy involves selecting the action that results in the highest estimate of the controller's value function, $q(s, g, a; w)$. Actions continue to be selected by the controller while

an internal critic monitors the current state, comparing it to the current subgoal, and delivering an appropriate *intrinsic reward*, $\tilde{r}$, to the controller on each time step. Each transition experience, $(s, g, a, \tilde{r}, s')$, is recorded in the controller's experience memory set, $\mathcal{D}_1$, to support learning. When the subgoal is attained, or a maximum amount of time has passed, the meta-controller observes the resulting state, $s_{t'} = s_{t+T+1}$, and selects another subgoal, $g' = g_{t+T+1}$, at which time the process repeats, but not before recording a transition experience for the meta-controller, $(s, g, G, s_{t'})$ in the meta-controller's experience memory set, $\mathcal{D}_2$. The parameters of the value function approximators are adjusted based on the collections of recent experiences. For training the meta-controller value function, we minimize a loss function based on the reward received from the environment:

$$\mathcal{L}_i(\mathcal{W}) \triangleq \mathbb{E}_{(s,g,G,s_{t'}) \sim \mathcal{D}_2} \big[ \big( \mathcal{Y} - Q(s, g; \mathcal{W}) \big)^2 \big], \tag{1}$$

where $G = \sum_{t'=t}^{t+T} \gamma^{t'-t} r_{t'}$ is the accumulated external reward (return) between the selection of consecutive subgoals. $\mathcal{Y} = G + \gamma \max_{g'} Q(s', g'; \mathcal{W})$ is the target value for the expected return at the time that the meta-controller selected subgoal $g$. The controller improves its subpolicy, $\pi(a|s, g)$, by learning its value function, $q(s, g, a; w)$, over the set of recorded transition experiences. The controller updates its value function approximator parameters, $w$, so as to minimize its loss function:

$$L_i(w) \triangleq \mathbb{E}_{(s,g,a,\tilde{r},s') \sim \mathcal{D}_1} \big[ \big( y - q(s, g, a; w) \big)^2 \big], \tag{2}$$

where $y = \tilde{r} + \gamma \max_a' q(s', g, a'; w)$ is the target expected intrinsic return value.

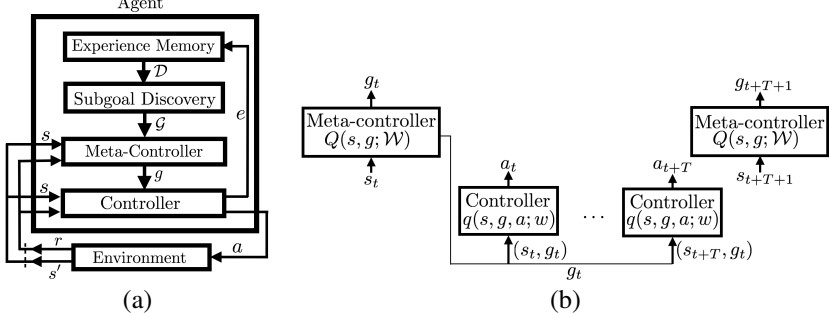

Figure 1: (a) The information flow in the unified Model-Free Hierarchical Reinforcement Learning Framework. (b) Temporal abstraction in the meta-controller/controller framework.

## 3.2 INTRINSIC MOTIVATION LEARNING

The acquisition of hierarchies of reusable skills is one of the distinguishing characteristics of biological intelligence, and the learning of such hierarchies is an important open problem in computational reinforcement learning. In humans, these skills are learned during a substantial developmental period in which individuals are intrinsically motivated to explore their environment and learn about the effects of their actions (Vigorito & Barto, 2010).

Intrinsic motivation learning is the core idea behind the learning of value functions in the meta-controller and the controller. In some tasks with sparse delayed feedback, a standard RL agent cannot effectively explore the state space so as to have a sufficient number of rewarding experiences to learn how to maximize rewards. In contrast, the intrinsic critic in our HRL framework can send much more regular feedback to the controller, since it is based on attaining subgoals, rather than ultimate goals. As an example, our implementation typically awards an intrinsic reward of $+1$ when the agent attains the current subgoal, $g$, and $-1$ for any other state transition. Successfully solving a difficult task not only depends on such an intrinsic motivation learning mechanism, but also on the meta-controller's ability to learn how to choose the right subgoal for any given state, $s$, selecting the subgoals from a set of candidate subgoals. Indeed, identifying a good set of candidate subgoals is an additional prerequisite for success.

## 3.3 UNSUPERVISED SUBGOAL DISCOVERY

The performance of the meta-controller/controller framework depends critically on selecting good candidate subgoals for the meta-controller to consider.

What is a subgoal? In our framework, a subgoal is a state, or a set of closely related states, that satisfies at least one of these conditions:

1. It is close (in terms of actions) to a rewarding state. For example, in the rooms task in Figure 2(a), the key and lock are rewarding states.

2. It represents a set of states, at least some of which tend to be along a state transition path to a rewarding state.

For example, in the rooms task, the red room should be visited to move from the purple room to the blue room in order to pick up the key. Thus any state in the red room is a reasonably good subgoal for an agent currently in the purple room. Similarly, the states in the blue room are all reasonably good subgoals for an agent currently in the red room. The doorways between rooms can also be considered as good subgoals, since entering these states allows for the transition to a set of states that may be closer to rewarding states.

Our strategy involves leveraging the set of recent transition experiences that must be recorded for value function learning, regardless. Unsupervised learning methods applied to sets of experiences can be used to identify sets of states that may be good subgoal candidates. We focus specifically on two kinds of analysis that can be performed on the set of transition experiences. We hypothesize that good subgoals might be found by (1) attending to the states associated with *anomalous* transition experiences and (2) clustering experiences based on a similarity measure and collecting the set of associated states into a potential subgoal. Thus, our proposed method for subgoal discovery merges *anomaly (outlier) detection* with online $K$-means clustering of experiences.

### 3.3.1 ANOMALY DETECTION

The anomaly (outlier) detection process identifies states associated with experiences that differ significantly from the others. In the context of subgoal discovery, a relevant anomalous experience would be one that includes a substantial positive reward in an environment in which reward is sparse. We propose that the states associated with these experiences make for good candidate subgoals. For example, in the rooms task, transitions that arrive at the key or the lock are quite dissimilar to most transitions, due to the large positive reward that is received at that point.

Since the goal of RL is maximizing accumulated (discounted) rewards, these anomalous experiences, involving large rewards, are ideal as subgoal candidates. Large changes in state features can also be marked as anomalous. In some computer games, like the ATARI 2600 game *Montezuma's Revenge*, each screen represents a room, and the screen changes quickly when the agent moves from one room to another. This produces a large distance between two consecutive states. Such a transition can be recognized simply by the large instantaneous change in state features, marking the associated states as reasonable candidate subgoals. There is a large literature on anomaly detection, in general, offering methods for applying this insight. Heuristic meta-parameter thresholds can be used to identify dissimilarities that warrant special attention, or unsupervised machine learning methods can be used to model the joint probability distribution of state variables, with low probability states seen as anomalous.

### 3.3.2 K-MEANS CLUSTERING

The idea behind using a clustering algorithm is "spatial" state space abstraction and dimensionality reduction with regard to the internal representations of states. If a collection of transition experiences are very similar to each other, this might suggest that the associated states are all roughly equally good as subgoals. Thus, rather than considering all of those states, the learning process might be made faster by considering a representative state (or smaller set of states), such as the centroid of a cluster, as a subgoal. Furthermore, using a simple clustering technique like $K$-means clustering to find a small number of centroids in the space of experiences is likely to produce centroid subgoals that are dissimilar from each other. Since rewards are sparse, this dissimilarity will be dominated by state features. For example, in the rooms task, the centroids of the $K$-means clustering with $K = 4$ lie close to the geometric center of each room, with the states within each room coming to belong to the corresponding subgoal's cluster. In this way, the clustering of transition experiences can approximately produce a coarser representation of state space, in this case replacing the fine grained "grid square location" with the coarser "room location".

### 3.4 UNIFYING UNSUPERVISED LEARNING WITH MODEL-FREE HRL

These conceptual components can be unified into a single model-free HRL framework. The major components of this framework and the information flow between these components are schematically displayed in Figure 1(a). At time $t$, the meta-controller observes the state, $s = s_t$, from the environment and chooses a subgoal, $g = g_t$, either from the discovered subgoals or from a random set of states (to promote exploration). The controller receives an input tuple, $(s, g)$, and is expected to learn to implement a subpolicy, $\pi(a|s, g)$, that solves the *subtask* of reaching from $s$ to $g$. The controller selects an action, $a$, based on its policy, in our case directly derived from its value function, $q(s, g, a; w)$. After one step, the environment updates the state to $s'$ and sends a reward $r$. The transition experience $(s, g, a, \tilde{r}, s')$ is then stored in the experience memory for the controller, $\mathcal{D}_1$. If the internal critic detects that the resulting state, $s'$, is the current goal, $g$, the experience $(s_t, g, G, s_{t'})$ is stored in the meta-controller experience memory, $\mathcal{D}_2$, where $s_t$ is the state that prompted the selection of the current subgoal, and, $s_{t'} = s_{t+T}$ is the state when meta-controller assigns the next subgoal $g' = g_{t'}$. The experience memory sets are typically used to train the value function approximators for the meta-controller and the controller by sampling a random minibatch of recent experiences. The subgoal discovery mechanism exploits the underlying structure in the experience memory sets using unsupervised anomaly detection and experience clustering. A detailed description of this process is outlined in Algorithm 1.

## 4 EXPERIMENTS

We conducted numerical experiments in order to investigate the ability of the unsupervised subgoal discovery method in discovering useful subgoals, as well as the efficiency of the unified model-free hierarchical reinforcement learning framework in learning robust representations. The simulations were conducted in two environments with sparse delayed feedback: a variant of the rooms task in Figure 2 and a classic ATARI 2600 game called "Montezumas Revenge".

### 4.1 4-ROOMS TASK WITH KEY AND LOCK

Consider the task of navigation in the *4-rooms environment with a key and a lock* as shown in Figure 2(a). At the beginning of each episode, the agent was initialized in an arbitrary location in an arbitrary room. The agent received $r = +10$ reward for reaching the key and $r = +40$ if it moved to the box while carrying the key. The agent could move either $\mathcal{A} = \{North, \ South, \ East, \ West\}$ on each time step. Bumping into the wall boundaries was punished with a reward of $r = -2$. There was no reward for exploring the space. Learning in this environment with sparse delayed feedback is challenging for a reinforcement learning agent. To successfully solve the task, the agent should represent knowledge at multiple levels of spatial and temporal abstractions. The agent should also learn to explore the environment efficiently.

First, we conducted an experiment based on a random walk. The agent was allowed to explore the *rooms* environment for 10,000 episodes. Each episode ended either when the rooms task was completed or after reaching maximum steps 200. We collected the agent's recent experiences $e = (s, a, s', r)$ in an experience memory $\mathcal{D}$. The stream of external rewards for each transition was used for detecting the *anomalous* subgoals (see Figure 2(c)). We applied a heuristic anomaly detection method for the streaming rewards that was able to differentiate between the rare large rewards and the regular small ones. These peaks, as shown in Figure 2(c), correspond to the experiences in which the key was reached ($r = +10$), or the experience of reaching to the lock after holding the key. Then, we implemented an incremental $K$-means clustering to the experience memory (See Algorithm 1). The centroids of the $K$-means clusters with $K = 4$ is plotted in Figure 2(b). After a few iterations, the centroid of clusters were interestingly found in the center of the rooms.

After the successful unsupervised subgoal discovery, we attempted to train a meta-controller and a controller using the discovered subgoals $\mathcal{G}$ (see Algorithm 1). We used a total of 6 subgoals, 2 anomalous ones and 4 centroids. The value function approximators were implemented as multi-layer artificial neural networks augmented to encourage the learning of sparse internal representations of states. The controller network, $q(s, g, a; w)$, takes the state, $s$, and the goal, $g$, as inputs. The Gaussian representation was used for preprocessing $s$ to input the network.

---

**Algorithm 1** Unified Model-Free Hierarchical Reinforcement Learning Algorithm

---

   **Initialize** discovered subgoals set $\mathcal{G} \leftarrow \emptyset$
   **Initialize** controller and meta-controller experience memories $\mathcal{D}_1$ and $\mathcal{D}_2$
   **Initialize** parameters of $q(.,.,.;w)$ and $\mathcal{Q}(.,.;\mathcal{W})$ randomly
   **Inputs:** learning rate $\alpha$, exploration rate $\epsilon$
   **for** episode $= 1, \ldots, M$ **do**
      Initialize state $s_0 \in \mathcal{S}$, $s \leftarrow s_0$, episode return $G \leftarrow 0$
      **if** Phase I of learning (before diescovering subgoals) **then**
         Choose a subgoal $g$ randomly form $\mathcal{S}$
      **else**
         Compute $Q(s, g; \mathcal{W})$
         $g \leftarrow \texttt{EPS-GREEDY}(Q(s, g; \mathcal{W}), \epsilon)$
      **end if**
      **Intrinsic Motivation Learning Algorithm:**
      **repeat** for each step $t = 1, \ldots, T$
         compute $q(s, g, a; w)$
         $a \leftarrow \texttt{EPS-GREEDY}q(s, g, a; w), \epsilon)$
         Take action $a$
         Observe next state $s'$ and external reward $r$
         Compute intrinsic reward $\tilde{r}$ from internal critic:
         $\tilde{r} \leftarrow +1$ if $s'$ reached to $g$ and, $\tilde{r} \leftarrow \min(-1, r)$ otherwise
         Store controller's transition experience $e = \{s, g, a, \tilde{r}, s'\}$ to $\mathcal{D}_1$
         Sample random minibatch of transitions $J_1 \subset \mathcal{D}_1$ and $J_2 \subset \mathcal{D}_2$
         Compute gradients of losses for controller, $\nabla L$ and meta-controller, $\nabla \mathcal{L}$
         Update controller's parameters $w$ and meta-controller's parameters $\mathcal{W}$:
         $w \leftarrow w - \alpha \nabla L, \quad \mathcal{W} \leftarrow w - \alpha \nabla \mathcal{L},$
         $s \leftarrow s', \quad G \leftarrow G + r$
         Anneal exploration rate $\epsilon$
      **until** $s$ is terminal or intrinsic task is done
      Store meta-controller's transition experience $e = \{s_0, g, G, s'\}$ to $\mathcal{D}_2$
      **Unsupervised Subgoal Discovery Algorithm:**
      **for** each $e = \{s, a, r, s'\} \in \mathcal{D}_1$ **do**
         **if** experience $e$ is an outlier (anomoly) **then**
            Store $s'$ to the subgoals set $\mathcal{G}$
            Remove $e$ from $\mathcal{D}$
         **end if**
      **end for**
      Fit a $K$-means Clustering Algorithm on $\mathcal{D}$
      Store the centroids to the subgoals set $\mathcal{G}$
   **end for**

---

The $k$-Winners-Take-All mechanism ($k$WTA) was used to produce a sparse conjunctive representation for the state, $s$ by only letting $10\%$ of the hidden units have high activations (O'Reilly & Munakata, 2001; Rafati & Noelle, 2015). The subgoals layer was connected only to the corresponding row of hidden units. This mechanism was included in hopes of avoiding catastrophic interference during training. The hidden layer is connected fully to the output units.

The training consisted of two phases. In the first phase, the controller was trained to navigate from any arbitrary state $s$ to reach the corresponding subgoal input $g$ from the subgoals set for a total of 100,000 episodes. When a centroid was selected as a subgoal, if the agent entered the corresponding cluster, the subgoal was considered attained. Thus, the controller essentially learned how to navigate from any location to any clusters (*room*) and also to any anomalous subgoals (key and door). The learning rate was $\alpha = 0.001$, the discount factor was $\gamma = 0.99$, and the exploration rate was set to $\epsilon = 0.2$. The average success rate (over 10 consecutive episodes) for the first phase of intrinsic motivation learning is shown in Figure 2(d). In the second phase of learning, we trained both meta-controller and controller for 100,000 episodes. The meta-controller's network consisted of two layers. The first layer, was a one-hot encoding of the state computed by converting the state to the index of the corresponding subgoal (cluster index or anomalous states) connected directly to the

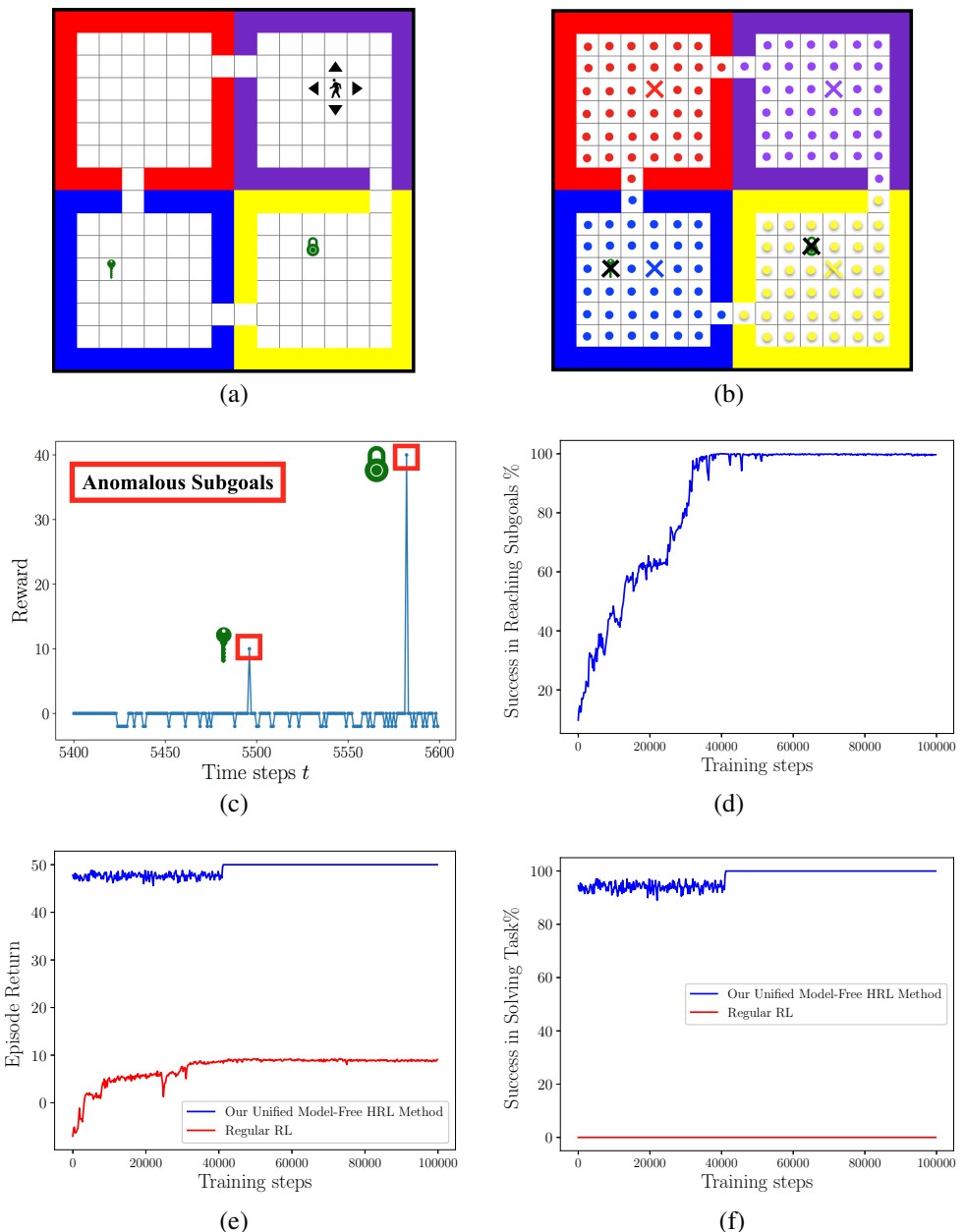

Figure 2: (a) The *rooms* task with a key and a box. (b) The results of the unsupervised subgoal discovery algorithm with *anomalies* marked with black Xs and *centroids* with colored ones. (c) Reward over an episode, with anomalous points corresponding to the key ($r = +10$) and the car ($r = +40$). (d) The average success of the controller in reaching subgoals during intrinsic motivation learning in the pretraining phase. (e) The average episode return for the unified model-free HRL method. (f) The success of the unified model-free HRL in solving the rooms task.

output layer. The average return (over 10 consecutive episodes) of the training episodes is shown in Figure 2(e). The agent learned very fast to converge the optimal policies by collecting maximum +50 rewards. The exploration rate was $\epsilon = 0.2$ and although this caused high stochasticity, the meta-controller and controller could robustly solve the task more than $90\%$ of the time and after about 40,000 episodes, the success rate was $100\%$ as shown in Figure 2(f). We also compared the efficiency of learning representations in our unified HRL method with the results from training a network with regular RL algorithms (TD SARSA method). The function approximator that we used

for $Q(s, a; W)$ was similar to the controller, and we used the same values for the training hyper parameters. Surprisingly, the regular RL could only reach the key before becoming stuck in that region due to the high local reward. Although we used $20\%$ exploration rate, the agent was not motivated to explore the rest of the state space to reach the box and solve the task as shown in Figure 2(e) and (f) (red plots).

## 4.2 MONTEZUMA'S REVENGE

The first room of *Montezuma's Revenge* (see Figure 3(a) is considered. The game is well-known to be a challenging task for agents because the game requires solving many subtasks and avoiding traps. Most challenging of all is the sparse delayed rewards feedback. The agent should learn to navigate the *man* in red in order to reach the *key* by the sequence of: (1) passing the *middle ladder* (2) passing the *bottom right ladder* (3) passing the *bottom left ladder* (4) reaching the key (5) After picking up the key ($r = +100$), it should return back in the reverse sequence and attempt to reach the *door* ($r = +300$) and exit the room. The moving skull makes reaching to the key extremely strenuous, thus the random walk in this environment faces early failures. The agent requires the intrinsic motivation learning in order to explore the environment in a more efficient way (Kulkarni et al., 2016). The DeepMind's Deep Q-Learning (DQN) algorithm (Mnih et al., 2015) which successfully surpassed the human expert in most of the ATARI 2600 games, failed to learn this game since the agent did not reach any rewarding state in the initial exploration stage.

Inspired by the work of Kulkarni et al. (2016), we used a two-stage meta-controller/controller framework (Figure 1(b)) to learn the representations of the temporal abstractions. The meta-controller and controller were trained in two phases. In the first phase of training, the controller was trained to navigate the man from any location in the given frame $s$ to any other location specified in a subgoal frame $g$. An initial set of random locations based on a custom edge detection algorithm was used for pretraining the controller through intrinsic motivation. Unsupervised object detection in the scene using state of the art computer vision algorithms is a challenging task (Kulkarni et al., 2016; Fra, 2015), but we hypothesized that in most of the games the location of edges where the object is separated from their background can be good initial candidates. We used a variant of the DQN deep Convolutional Neural Network (CNN) architecture (Figure 3(b)) for function approximation of the controller's state-action value function $q(s, g, a; w)$. The input to the controller consisted of four consecutive frames of size $84 \times 84$ as state $s$, and an additional frame binary mask of the subgoal $g$ attained from the aforementioned random subgoals set. The concatenated state and subgoal frames $(s, g)$, were passed to $q(s, g, a; w)$. The controller then took action from 18 different joystick choices based on a policy derived from $q(s, g, a; w)$.

During the intrinsic motivation learning, the recent experiences were saved in a experience memory $\mathcal{D}$ with size of $10^6$. In order to compare our results with the existing methods, we used the learning parameters of Mnih et al. (2015). For example, the learning rate was set to to be $2.5 \times 10^4$, with a discount rate of $\gamma = 0.99$. In the first phase, we only trained the network for a total of $2.5^6$ time steps. The epsilon decreased from $1.0$ to $0.1$ in the first one million steps of the training and remained fixed after that. Every $100,000$ steps, we applied the incremental unsupervised subgoal discovery to the experience memory to find the new subgoals in terms of anomalies and centroids. As shown in the Figure 3(d), the unsupervised learning algorithm managed to discover the location of the key and doors by the unsupervised anomaly detection algorithm and also the useful objects such as ladders, stages and the rope by applying the incremental $K$-means clustering with $K = 10$ to the set of locations of the man. In the second phase of learning after the successful subgoal discovery, we trained the meta-controller and the controller jointly. We used an architecture based on the DQN CNN (Mnih et al., 2015), as shown in Figure 3(c) for the meta-controller's value function $\mathcal{Q}(s, g; \mathcal{W})$. We used the non-overlapping discovered subgoals $\mathcal{G}$ which resulted in a set of 11 subgoals, $\mathcal{G}$. At the beginning of each episode, the meta-controller assigned a subgoal $g \in \mathcal{G}$ based on an epsilon-greedy policy derived from $\mathcal{Q}(s, g; \mathcal{W})$, after observing the four consecutive frames of the game. The controller then attempted to reach these subgoals. The meta-controller's experience memory, $\mathcal{D}_1$ had a size of $10^6$, and the meta-controller's memory size was $5 \times 10^4$. The cumulative rewards for the game episodes is shown in Figure 3(e). After about 1.5 million time steps, the controller managed to reach the key subgoal more frequently. The success of the intrinsic motivation learning is depicted in Figure 3(f). At the end of the second phase of learning, i.e. after 2.5 million learning steps, the meta-controller chose the proper subgoals that led to more frequent success in solving the first room and collecting the maximum rewards ($+400$).

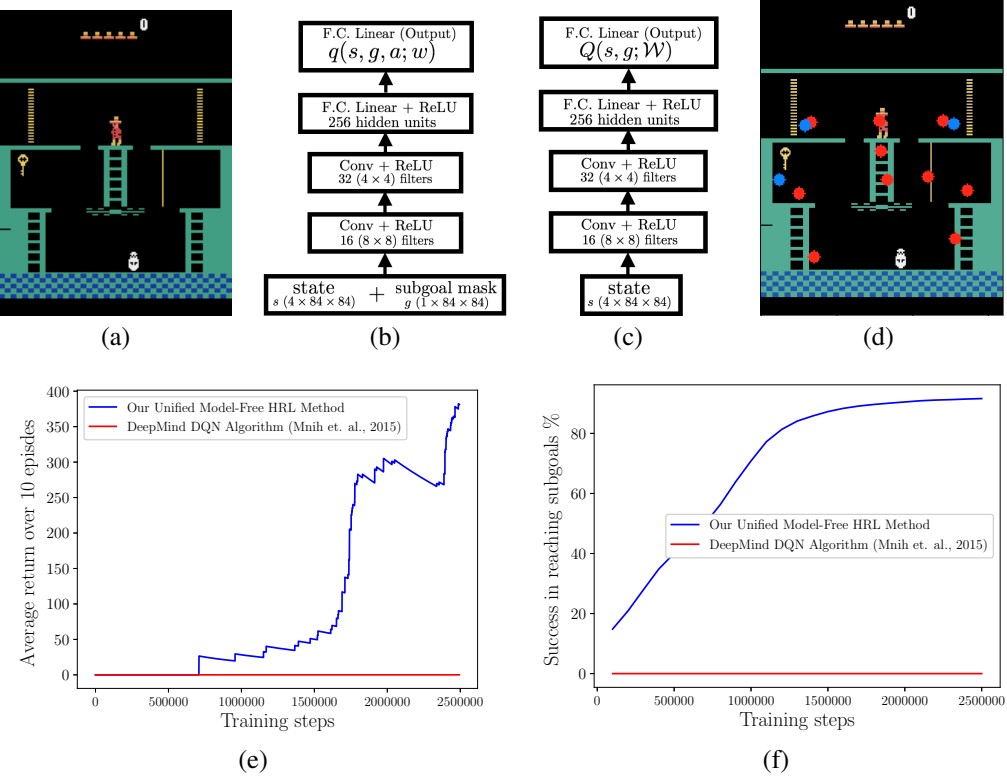

Figure 3: (a) A sample screen from the ATARI 2600 game *Montezuma's Revenge*. (b) The CNN architecture for the controller's value function. (c) The CNN architecture for the meta-controller's value function. (d) The results of the unsupervised subgoal discovery algorithm. The blue circles are the discovered anomalous subgoals and the red ones are the centroid subgoals. (e) The average of return over 10 episode during the second phase of the learning. (f) The success of the controller in reaching to the subgoals during the second phase of learning.

## 5 DISCUSSIONS

Research on human and animal behavior has long emphasized its hierarchical structure (Botvinick et al., 2009). There is empirical and computational evidence that suggests how temporal abstraction in the hierarchical reinforcement learning might map onto neural structures, in particular regions within the dorsolateral and orbital Prefrontal Cortex (PFC) (Botvinick et al., 2009; Badre et al., 2010). There are also studies on the interaction between the hippocampus and the PFC that is directly related to our unsupervised subgoal discovery method. Preston & Eichenbaum (2013) illustrated how novel memories (like *anomalous* subgoals) could be reinforced into permanent storage. Additionally, their studies suggest how PFC may play a major role in finding new meaningful representations from replaying the memory — a process akin to the clustering of recent experience memory $\mathcal{D}$ in our novel HRL framework. The latter mechanism is the essence of so-called *temporal abstraction* that furthermore helps with optimizing brain's large-scale memory retrieval processes.

## 6 CONCLUSION

We propose and implement a novel model-free method for subgoal discovery using incremental unsupervised learning over a small memory of the most recent experiences of the agent. When combined with an intrinsic motivation learning mechanism, this method learns subgoals and skills together, based on experiences in the environment. Thus, we offer an original approach to HRL that does not require the acquisition of a model of the environment, suitable for large-scale applications. We conducted experiments using our method on large-scale RL problems, such as portions of the difficult Atari 2600 game *Montezuma's Revenge*.

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
