# OpenReview forum: "Learning Representations in Model-Free Hierarchical Reinforcement Learning"
_ICLR.cc/2019/Conference_

### Official Review · AnonReviewer3 · 2018-11-02

**Rating:** 3
**Confidence:** 5

**Review:**

This paper proposes an unsupervised method for subgoal discovery and shows how to combine it with a model-free hierarchical reinforcement learning approach. The main idea behind the subgoal discovery approach is to first build up a buffer of “interesting” states using ideas from anomaly detection. The states in the buffer are then clustered and the centroids are taken to be the subgoal states.

Clarity:
I found the paper somewhat difficult to follow. The main issue is that the details of the algorithm are scattered throughout the paper with Algorithm 1 describing the method only at a very high level. For example, how does the algorithm determine that an agent has reached a goal? It’s not clear from the algorithm box. Some important details are also left out. The section on Montezuma’s Revenge mentioned that the goal set was initialized using a “custom edge detection algorithm”. What was the algorithm? Also, what exactly is being clustered (observations or network activations) and using what similarity measure? I can’t find it anywhere in the paper. Omissions like this make the method completely unreproducible.

Novelty:
The idea of using clustering to discover goals in reinforcement learning is quite old and the paper does a poor job of citing the most relevant prior work. For example, there is no mention of “Dynamic Abstraction in Reinforcement Learning via Clustering” by Mannor et al. or of “Learning Options in Reinforcement Learning” by Stolle and Precup (which uses bottleneck states as goals). The particular instantiation of clustering interesting states used in this paper does seem to be new but it is important to do a better job of citing relevant prior work and the overall novelty is still somewhat limited.

Significance:
I was not convinced that there are significant ideas or lessons to be taken away from this paper. The main motivation was to improve scalability of RL and HRL to large state spaces, but the experiments are on the four rooms domain and the first room of Montezuma’s Revenge, which is not particularly large scale. Existing HRL approaches, e.b. Feudal Networks from Vezhnevets et al. have been shown to work on a much wider range of domains. Further, it’s not clear how this method could address scalability issues. Repeated clustering could become expensive and it’s not clear how the number of clusters affects the approach as the complexity of the task increases. I would have liked to see some experiments showing how the performance changes for different numbers of clusters because setting the number of clusters to 4 in the four rooms task is a clear use of prior knowledge about the task.

Overall quality:
The proposed approach is based on a number of heuristics and is potentially brittle. Given that there are no ablation experiments looking at how different choices (number of clusters/goals, how outliers are selected, etc) I’m not sure what to take away from this paper. There are just too many seemingly arbitrary choices and moving parts that are not evaluated separately.

Minor comments:
- Can you back up the first sentence of the abstract? AlphaGo/AlphaZero do well on the game of Go which has ~10^170 valid states.
- First sentence of introduction. How can the RL problem have a scaling problem? Some RL methods might, but I don’t understand what it means for a problem to have scaling issues.
- Please check your usage of \cite and \citep. Some citations are in the wrong format.
- The Q-learning loss in section 2 is wrong. The parameters of the target (r+\gamma max Q) are held fixed in Q-learning.

---

### Official Review · AnonReviewer2 · 2018-11-02
**The methods seem somewhat tailored for the tasks and the results on the harder problem are not that convincing.**

**Rating:** 4
**Confidence:** 4

**Review:**

Summary:
The authors propose an HRL system which learns subgoals based on unsupervised analysis of recent trajectories. The subgoals are found via anomaly/outlier detection (in this case states with a very high reward) and the clustering together of states that are very similar. The system is evaluated on the 4-rooms task and on the atari game Montezuma’s Revenge.

The paper cites relevant work and provides a nice explanation of subgoal-based HRL. The paper is for the most part well-written and easy to follow.

The experiments are unfortunately not making a very convincing case for the general applicability of the the methods. While the system does not employ a model of the environment, k-means clustering based on distances seems to be particularly well-suited for the two environments investigated in the paper. It is known that the 4-rooms experiment is much easier to solve with subgoals that correspond to the rooms themselves. I can only conclude from this experiment that k-means can find those subgoals given the right number (4) of clusters and injecting the knowledge that distances in grid-worlds correlate well with transition probabilities. Similarly, the use of distance-based clustering seems well-suited for games with different rooms like Montezuma’s Revenge but that might not generalize to many other games.

The anomaly detection subgoal discovery is interesting as a method to speed-up learning but it still requires these (potentially sparse) high reward states to be found first. For tasks with sparse rewards it does make sense to set high reward states as potential subgoals instead of waiting for value to propagate. That said, the reward for the lower level policy is only less sparse in the sense that wasting time gets punished with a negative reward. Subgoal discovery based on rewards should probably also take the ability of the current policy to obtain those rewards into account like some other methods for subgoal discovery do (see for example Florensa et al., 2018). The authors mention that the subgoals were manually chosen by Kulkarni et al. (2016) instead of learned in an unsupervised way but I don’t think that the visual object detection method employed there is that much more problem specific.

Like Kulkarni et al. (2016), the authors compare their method with DQN (Mnih et al. 2015) but it was already known that that baseline cannot solve the task at all and a lot more results on Montezuma’s Revenge have been published since then. A more insightful baseline would have been to compare with at least some other HRL methods that are able to learn the task to some extend like perhaps Feudal Networks (Vezhnevets et al., 2017). Looking at the graph in the Feudal Networks paper for comparison, the results in this paper seem to be on par with the LSTM baseline there but it is hard to compare this on the basis of the number of episodes. Did the reward go up further after running the experiment longer?

Since the results are not that spectacular and a comparison with prior work is lacking, the main contributions of the paper are more conceptual. I think that it is interesting to think more carefully about how sparse reward states and state similarities can be used more efficiently but the ideas in the paper are not original or theoretically founded enough to have a lot of impact without the company of stronger empirical results.

Extra reference:
Carlos Florensa, David Held, Xinyang Geng, Pieter Abbeel. (2017). Automatic goal generation for reinforcement learning agents. arXiv preprint arXiv:1705.06366.

---

### Official Review · AnonReviewer1 · 2018-11-03
**A simple ideas works well in a challenging RL domain. The generalizability and significance can be improved if more domains can be tested**

**Rating:** 5
**Confidence:** 4

**Review:**

This paper proposed a model-free HRL method, which is combined with unsupervised learning methods, including abnormality discovery and clustering for subgoal discovery. In all, this paper studies a very important problem in RL and is easy to follow. The technique is sound. Although the novelty is not that significant (combining existing techniques), it showed good results on Montezuma’ revenge, which is considered as a very challenging  problem for primitive action based RL.

Although the results are impressive, I still have some doubt about the generalizability of the method. It might be helpful to improve its significance if more diversified domains can be tested.

The paper can be strengthen by providing some ablation test, for example, is performance under different K for Kmeans?

Also some important details seems missing, for example, the data used for kmeans, it is mentioned that the input to the controller is four consecutive frame of size 84x84, so the input data dimension is more than 10k, I guess some further dimensionality reduction technique has to be applied in order to run kmeans effectively.

Regarding the comparisons, the proposed method is only compared with one primitive action based method. It might be better to include results from other HRL methods, such as Kulkarni et al.

Is the curve based on the mean of different runs? It might be useful to include an errorbar to show the statistical significance.

---

### Meta-Review · Area_Chair1 · 2018-12-14

**Confidence:** 5
**Recommendation:** Reject

**Metareview:**

Pros:
- good results on Montezuma

Cons:
- moderate novelty
- questionable generalization
- lack of ablations and analysis
- lack of stronger baselines
- no rebuttal


The reviewers agree that the paper should be rejected in its current form, and the authors have not bothered revising it to take into account the detailed reviews.